# Chronic-Exposure Low-Frequency Magnetic Fields (Magnetotherapy and Magnetic Stimulation) Influence Serum Serotonin Concentrations in Patients with Low Back Pain—Clinical Observation Study

**DOI:** 10.3390/ijerph19159743

**Published:** 2022-08-08

**Authors:** Marta Woldańska-Okońska, Kamil Koszela

**Affiliations:** 1Department of Rehabilitation and Physical Medicine, Medical University, 90-700 Łódź, Poland; 2Neuroorthopedics and Neurology Clinic and Polyclinic, National Institute of Geriatrics, Rheumatology and Rehabilitation, 02-637 Warsaw, Poland

**Keywords:** LBP, musculoskeletal disorders, spine, conservative treatment, rehabilitation, magnetotherapy effects, physical therapy

## Abstract

(1) Background: The influence of serotonin on many regulatory mechanisms has not been sufficiently studied. The use of a physical method, assuming the possibility of its action on increasing the concentration of serotonin, may be the direction of therapy limiting the number of antidepressants used. The aim of the research was to study the effects of low-frequency magnetic fields of different characteristics on the circadian profile of serotonin in men with low back pain. (2) Methods: 16 men with back pain syndrome participated in the study. The patients were divided into two groups. In group 1, magnetotherapy (2.9 mT, 40 Hz, square wave, bipolar) was applied at 10.00 a.m. In group 2, the M2P2 magnetic stimulation program of the Viofor JPS device was used. Treatments in each group lasted 3 weeks, 5 days each, with breaks for Saturday and Sunday. The daily serotonin profile was determined the day before the exposure and the day after the last treatment. Blood samples (at night with red light) were collected at 8:00, 12:00, 16:00, 24:00, and 4:00. The patients did not suffer from any chronic or acute disease and were not taking any medications. (3) Results: In group 1, a significant increase in serotonin concentration was observed after 15 treatments at 4:00. In group 2, a significant increase in serotonin concentration was observed at 8:00 after the end of the treatments. In comparison between magnetotherapy and magnetic stimulation, the time points at which differences appeared after the application of serotonin occurred due to the increase in its concentrations after the application of magnetic stimulation. (4) Conclusions: Magnetotherapy and magnetic stimulation, acting in a similar way, increase the concentration of serotonin. Weak magnetic fields work similarly to the stronger ones used in TMS. It is possible to use them in the treatment of mental disorders or other diseases with low serotonin concentrations.

## 1. Introduction

Serotonin (5-hydroxytryptamine), similarly to melatonin, is produced in a circadian cycle. Its precursor is tryptophan; out of which, hydroxytryptophan is synthesized by the tryptophan hydroxylase (TPH) enzyme by means of hydroxylation. After decarboxylation, hydroxytryptophan is converted into serotonin [1,2,3].

Serotonin is mainly produced by intestinal enterochromaffin cells. Its small quantities are also synthesized in the central nervous system, rectum (tubular epithelial cells), bronchial epithelium, thyroid gland cells, ovaries, thymus, pancreas, breasts, and skin [2,4].

In rats, mast cells are an important source of serotonin. When it is released into the blood, serotonin is intercepted by platelets, and stored until it is released or broken down by monoamine oxidase (MAO). Serotonin is metabolized by liver or endothelial cells in the lungs, initially by means of deamination by MAO into 5-hydroxyindolacetaldehyde. From this form, it is transformed in the process of oxidation into 5-hydroxyindoleacetic acid (5-HIAA) [2].

Through the action of NAT and HIOMT enzymes in the brain, serotonin is converted to melatonin, and then, 6-hydroxymelatonin is produced in subsequent reactions [1]. Serotonin is involved in cognitive processes, and regulates eating habits, sexual activity, mood and anxiety [4], aggression and pain (through substance P inter alia), sleep and wakefulness, and other biological rhythms. It acts as a hormone and a cytokine [2,3,5]. All of these processes are associated with membrane receptors that are classified into seven families (5HT1-7) with 21 subtypes. These receptors, with the exception of 5HT3, bind with G protein [2].

Serotonin is a biogenic amine best known as a neurotransmitter. Serotonin has been used in animal models, which have been implemented as a way to study its role in humans. Through these studies, it has become clear that serotonin regulates gastrointestinal motility, peripheral vascular tone, cerebral vascular tone, and platelet function, and plays a role in the pathophysiology of mood disorders, vomiting, migraine, irritable bowel syndrome (IBS), and pulmonary and systemic hypertension. This knowledge is applied in animals in research on drugs that affect the serotonergic system. The increasing use and availability of drugs governing the serotonergic system created the circumstances that led to the discovery of new toxicity to both humans and animals. Serotonin syndrome has been described as the clinical picture seen in humans and animals of serotonin toxicity [6,7,8].

Serotonin has a positive effect on the secretion of hormones such as prolactin (PRL, CRF, ACTH, and cortisol) and the growth hormone (GH, LH, and FSH). It inhibits the synthesis of progesterone, and, thus, induces premenstrual syndrome (PMS). 5-hydroxytryptamine also affects the pituitary–thyroid axis, stimulating the secretion of TSH [9,10].

The concentration of serotonin rises in preeclampsia as it is released from the breakdown of platelets [3]. Marktl et al. [11] observed a decrease in the concentration of serotonin in patients with ischemic heart disease during the day, whereas its concentrations remained steady at night.

In peripheral tissues, serotonin plays a role in immune processes (proinflammatory agent) [3], the physiology of digestion, the development of mammary glands, coagulation, and fibrinolysis. In stress, by affecting the adrenal cortex, serotonin regulates cell proliferation, and their migration and differentiation [2].

Only a small percentage of the body’s serotonin (about 5%) can be found in the mature brain of mammals. In the intestine, pheochromocytomas are dispersed in the visceral epithelium from the stomach to the colon, and produce more than 95% of the serotonin in the human body [12].

In the embryonic period, serotonin takes part in the formation of nerve pathways, affects the formation of glial cells [13], and has a trophic effect on the development of the cerebral cortex [14]. Serotonin deficiency results in cognitive deficits. An excess of serotonin causes the inhibition of development of the sensorimotor cortex. It is also believed that an excess of serotonin may lead to autism [13].

The anatomical and neurochemical connection of the serotonergic system to the areas of the brain that regulate memory and learning have led current drug discovery programs to focus on this system as the primary therapeutic target of drugs. However, none of these programs have provided data to suggest that any of the new treatments have Alzheimer’s-disease-modifying properties [15]. 5-HT is a significant factor in linking disturbance of the circadian system and mood. Finding out how these two systems work together could provide new ways of treatment for depression [16].

In the skin, serotonin affects cell proliferation and the stress response system (serotonin/melatonin pathway) [2]. It also takes part in thermoregulation [3].

The knowledge of the effects of serotonin may explain the pathogenesis of a large number of diseases, and may facilitate treatment of many of them through its effects on 5HT1-7 receptors [17].

The recognition of the effects of magnetic fields on the concentration of serotonin may lead to therapeutic applications of them in numerous diseases that are associated with serotonin metabolism.

Due to the association of the level of pain with the occurrence of depression and its intensity, as well as its influence through various mechanisms of action, including the role of serotonin, antidepressants are recommended in the treatment of back pain syndromes. These drugs are the most frequently prescribed drugs in the UK. More doctors prescribe antidepressants (7.2 million) than opioid painkillers (5.6 million). Antidepressants are the fourth most frequently prescribed type of medication for low back pain in the United States. More than a quarter of Americans with chronic low back pain are prescribed an anti-depressant within three months of the first visit to their doctor. In the UK, 16% of prescriptions for antidepressants for children and adolescents are related to the treatment of pain [7].

Antidepressants are recommended by most (75%) of the guidelines for clinical standards for low back pain, where the American College of Physicians recommends the use of duloxetine with a serotonin norepinephrine reuptake inhibitor (SNRI). The UK’s National Institute of Health and Care Excellence recommends amitriptyline or duloxetine as the choice of treatment for people with various forms of neuropathic pain, and even back pain with radicular symptoms. The use of a physical method, assuming the possibility of its action on increasing the concentration of serotonin, may be the direction of therapy limiting the number of antidepressants used.

The aim of the research was to study the effects of low-frequency magnetic fields of different characteristics on the circadian profile of serotonin in men with low back pain.

Hypothesis: Magnetotherapy and magnetic stimulation have an influence on serum serotonin concentrations in patients with low back pain.

## 2. Materials and Methods

### 2.1. Study Population

The study was conducted in 16 men with low back pain syndrome that were admitted to the Rehabilitation Ward of the Regional Hospital in Sieradz, Poland. The study was approved by the Bioethics Committee for Scientific Research at Medical University in Lodz, number RNN/254/05/KB.

Inclusion criteria:-Men aged 18–60 years with LBP;-The men did not suffer from any diseases of the digestive system, circulatory system, metabolism, or hormonal disorders;-Not taking any medications;-No contraindications to the therapy from other systems;-Consent to examination procedures;

Exclusion criteria:


-Contraindications to the use of a magnetic field;-Lack of consent of the patient and the guardian to examinations and participation in the program.


### 2.2. Study Protocol—Magnetic Fields

Low-frequency magnetic fields used in physiotherapy provide only non-thermal effects, such as improving the tissue utilization of oxygen, anti-inflammatory, anti-edema, analgesic, angiogenetic, vasodilating, and sedative effects, and they accelerate the healing of wounds and bone fractures, reduce blood pressure, and improve the rheological parameters of blood. They affect blood glucose and cholesterol levels. The results obtained depend on the parameters used. As a result of the magnetostimulation, the biological hysteresis effect is observed, i.e., the effect of the magnetic field persisting one month after the end of the application [18].

The main difference between the two therapies is the induction and frequency of the magnetic field used, as well as the waveform. In the case of magnetostimulation, much lower induction values and a higher field frequency, even up to 3000 Hz, are used. However, the envelopes of the sent signals with a lower frequency, even a few Hz, generating ionic cyclotron resonance, have an active influence. Magnetotherapy, on the other hand, is a procedure that uses higher induction values, usually up to 40 mT, and much lower frequencies, 1–100 Hz. In magnetostimulation, the waveform of magnetic fields is called a sawtooth shape, whereas in magnetotherapy, they take the following shapes: sinusoid, rectangle, or triangle.

The patients were divided into two groups: Group 1 consisted of 6 men, with a mean age of 42 (31–54). An alternating magnetic field (2.9 mT, 40 Hz, bipolar square wave) was applied at 10.00 a.m. The applicator, in the form of a coil, was placed on the low back. Group 2 consisted of 10 men, with a mean age of 47 (37–56), who were treated with the M2P2 program of the Viofor JPS device at 10.00 a.m. (25–80-microT, 200 Hz). The whole of the patient’s body was exposed to a heterogenous magnetic field in the lying position with a mat as the applicator.

Magnetic fields were applied in each of the groups for 3 weeks, 5 days a week, with a break for weekends. The circadian profile of serotonin was estimated the day before the exposure (as control) and the day after 15 applications. Additionally, in both the groups, serotonin concentration was estimated a month after all the applications. Each patient served as their own control.

### 2.3. Study Protocol—Blood Analysis

The blood samples were taken in both groups at 08:00, 12:00, 16:00, 20:00, 24:00, and at 04:00. The hours well reflect the circadian rhythm of serotonin secretion, and exclude the effects of the diet on the concentration of the hormone in the serum, as the samples were taken on an empty stomach or before eating. At night, the samples were taken in red light. The patients remained in such light from 22:00 to 06:00 the day before and during the night that the samples were collected. The study on group 1 was conducted in early spring, and on group 2, in late autumn.

The serum that was obtained in centrifugation was stored in −20 °C until the concentration of serotonin was estimated. The concentrations were determined by radioimmunoassay (DRG, GmbH, Marburg, Germany).

### 2.4. Data Analysis

In order to answer the research questions, statistical analyses were performed with STATISTICA StatSoft Polska 2008.

The statistical analysis was carried out with the use of Student’s *t*-test for matched pairs, and the Wilcoxon signed-rank test with the significance level at *p* ≤ 0.05.

## 3. Results

Ideally, the daily serotonin concentration curve is the inversion of the daily serum melatonin concentration as a result of its metabolism. Whereas the melatonin concentration in the circadian cycle changes mainly with or without light, the serotonin concentration tends to decrease at night as the melatonin concentration increases. Nevertheless, there are many more factors influencing the serum concentration of serotonin than with melatonin; for example, eating meals. The curves below are less regular than the melatonin secretion curves, but they have a similar course, differing significantly at some time points after the application of magnetic fields in magnetotherapy and magnetic stimulation.

Figure 1 and Figure 2 show statistically significant differences after the application of magnetotherapy at 04:00 a.m., when an increase in serotonin concentration is observed, whereas the serotonin concentration is higher after magnetic stimulation at 08:00 a.m. These are the time points for low melatonin levels in circadian rhythm.

In Figure 3, before the applications of magnetic fields, the circadian curves of serotonin concentration differed statistically at 12:00, 16:00, and 04:00 h.

In Figure 4, after the applications of the magnetic fields, the circadian curves of serotonin differed significantly at all the time points. The time points at which differences appeared after the application of serotonin occurred due to the increase in its concentrations after the application of magnetic stimulation.

## 4. Discussion

The concentration of serotonin cannot be used as a measure of its production and secretion in the central nervous system due to the large number of sites for its synthesis. However, the serum levels of serotonin may reflect the extent to which it affects various systems and organs, resulting in the simultaneous expression of many types of receptors that are widely distributed throughout the central nervous system, endocrine glands, and tissues. Receptors determine the specific regulation of serotonin activity, including the analgesic effect [2]; hence, the use of antidepressants in low back pain.

The analgesic mechanism of antidepressants affecting the activity of serotonin in the central nervous system is complex. It is believed that they alleviate pain through multiple mechanisms [5]. These drugs reduce the severity of pain and raise the pain threshold, influencing the symptoms of depression and the vicious cycle of pain—anxiety–depression–insomnia. By affecting the neurotransmitter systems, antidepressants modulate nociceptive systems, inhibiting monoamine reuptake in the presynaptic membrane, enhancing serotonergic and noradrenergic transmission. Some of them also affect certain receptors of the serotonergic system.

Antidepressants influence unfavorable immune phenomena in depression, reducing the level of pro-inflammatory cytokines (e.g., TNF-α, IL-6), which are also pain mediators [7]. They also affect the structures of the central nervous system involved in the perception and assessment of pain by affecting the opioid and glutamatergic systems, and reduce the perception of pain stimuli.

The above mechanisms may explain [4,5] the action of physical stimuli in back pain syndromes, and the positive effect on melatonin levels in the diurnal cycle, which is related to the metabolism of serotonin [9].

Human clinical studies and experimental animal studies, in most cases, confirmed that magnetic fields have a positive effect on serum serotonin levels. Interestingly, different parameters of the magnetic fields had the same effect. The applied 25 Hz field in TMS (Transcranial Magnetic Stimulation) resulted in a higher concentration of serotonin in the frontal cortex of rats [19]. In a study by Keck et al. [20], TMS caused hormonal and behavioral changes, such as those found with antidepressants. Similarly, in this study, magnetic fields with different parameters caused an increase in serotonin concentration, although to an unequal degree in cases of different parameters.

A rise in the concentration of serotonin in the serum and in the pineal gland in mice and rats was caused by: the alternating magnetic field of 0.4 Gs constant induction [21], the constant magnetic field of low frequencies of 0.2 and 0.7 Gs (geomagnetic units) [22], and constant and alternating fields [10]. In the frontal cortex of rats, such a rise occurred after the application of the 1.8–3.8 mT, 10 Hz, sinusoidal wave magnetic fields [23].

In humans, an increase in the concentration of serotonin was observed after the application of the piko T—pT (10^−12T^) [24,25] magnetic fields and M2P2 magnetic stimulation [26]. The possible mechanism behind this phenomenon lies in the diminishing of MAO [27] and NAT activity [21,28]; an inhibition of beta-adrenergic activation of the pineal gland [11]; and, finally, a weakening of the blood–brain barrier, which may result in an increase in the concentration of tryptophan, the precursor of serotonin [29].

The foregoing observations have not been confirmed by only a handful of individual reports. No rise in the concentration of serotonin was observed in rats that were exposed for a long time to the 800 Gs fields of MRI (Magnetic Resonance Imaging) [30,31], where constant magnetic fields are predominant. The study was undertaken to assess the health risks in healthy patients and staff during the examination. It seems that the influence on increasing the level of serotonin is exerted by low-frequency variable magnetic fields with low induction, as in the above study.

In this study, an elevated concentration of serotonin was observed after magnetotherapy and magnetic stimulation (M2P2 program, which does not lower the nocturnal concentrations of melatonin) [32]. The consumed meals did not affect the changes in the concentration of serotonin, as the significant differences occurred at the times that were not related to food intake; that is, at 04:00 and 08:00, on an empty stomach.

This phenomenon may have an impact on an increased analgesic effect produced by magnetic fields [9]. Another positive effect occurs when weak magnetic fields are used in psychiatric therapies [24], where it is possible to avoid the adverse effects associated with the use of stronger fields of various parameters.

In the case of the physical therapy of back pain syndromes, the use of magnetic fields may help to reduce the use of painkillers and the increasingly wider use of antidepressants, which needs further studies [33,34].

One of the limitations of the study was the lack of a control group without any interventions, which could have added more information about the effectiveness of the interventions. However, the authors decided to treat all the participants because of the characteristics of the population. Additionally, the use of pain scales could show more correlation in the results. For financial reasons, the patients had to be treated at different times, in spring and autumn. If the patients were treated at the same time, the results could be more accurate.

The obtained results indicate that magnetotherapy and magnetic stimulation have an impact on the level of serotonin in men with low back pain; however, it requires further analyses supplemented with long-term follow-up in a larger group of patients in correlations of the levels of pain.

## 5. Conclusions

Working similarly, magnetotherapy and magnetic stimulation raise the concentrations of serotonin. However, it seems that magnetic stimulation is more effective in this respect.Weak magnetic fields produce similar effect to the stronger ones that are used in TMS.It is possible to use magnetotherapy and magnetic stimulation in the treatment of mental illnesses or in the diseases of the neuromuscular system, as well as in other illnesses that are characterized by a low concentration of serotonin.

## Figures and Tables

**Figure 1 ijerph-19-09743-f001:**
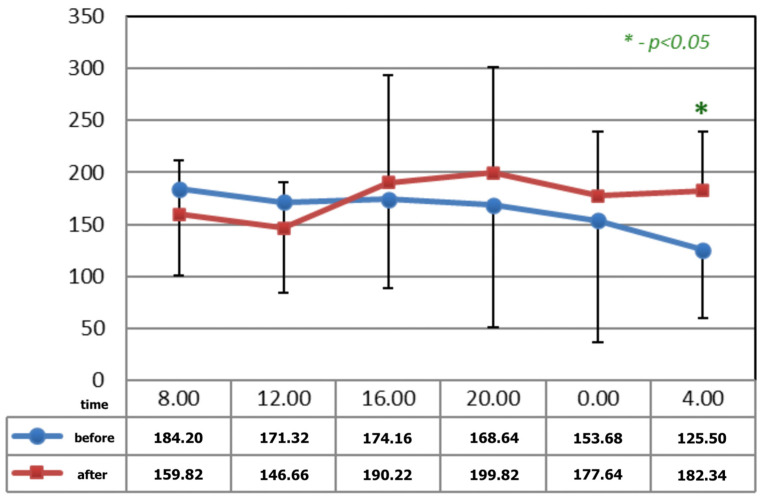
The circadian concentrations (ng/mL) of serotonin before and after 15 applications of magnetotherapy. * statistical significance.

**Figure 2 ijerph-19-09743-f002:**
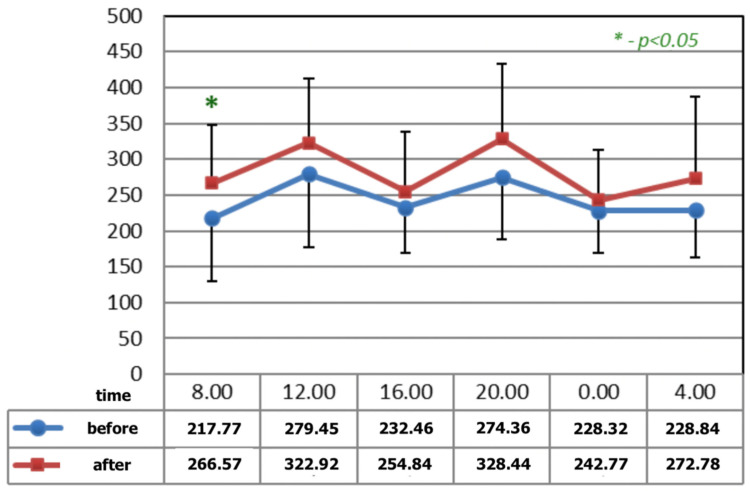
The circadian concentrations (ng/mL) of serotonin before and after the completion of M2 P2 magnetic stimulation. * statistical significance.

**Figure 3 ijerph-19-09743-f003:**
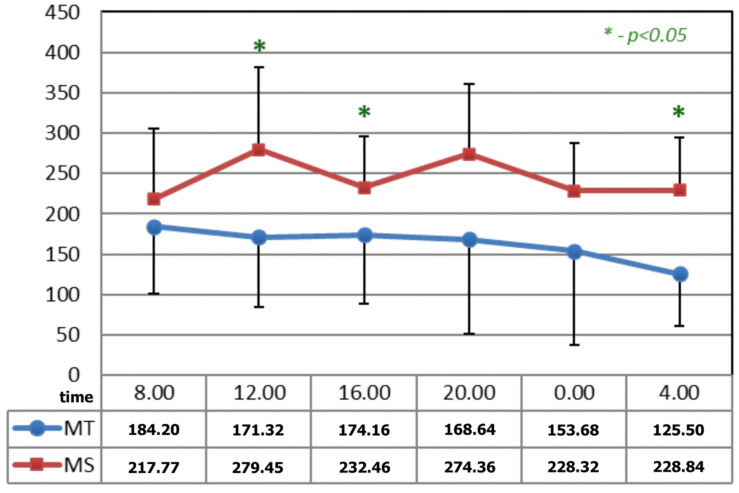
Comparison of the circadian concentration (ng/mL) of serotonin before the applications of magnetic fields—magnetotherapy and magnetic stimulation. * statistical significance.

**Figure 4 ijerph-19-09743-f004:**
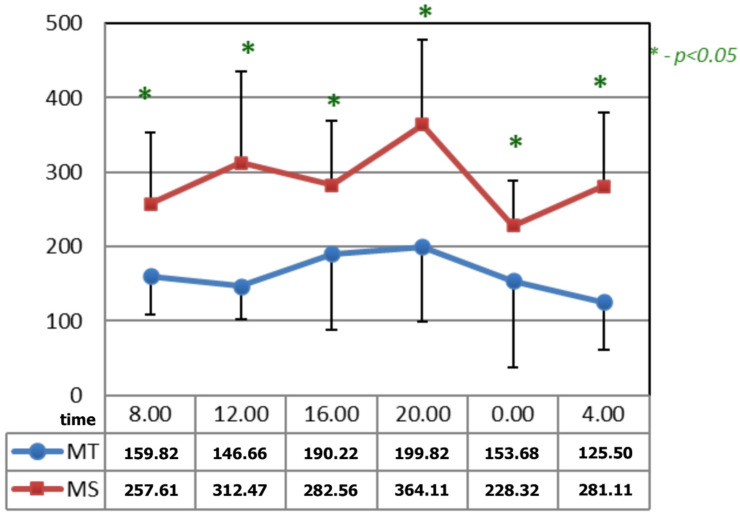
Comparison of the circadian concentration (ng/mL) of serotonin after 15 applications of magnetotherapy and magnetic stimulation. * statistical significance.

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
