# Peer review of "Chronic-Exposure Low-Frequency Magnetic Fields (Magnetotherapy and Magnetic Stimulation) Influence Serum Serotonin Concentrations in Patients with Low Back Pain—Clinical Observation Study"

_ijerph, 2022, doi:10.3390/ijerph19159743_

Round 1

Reviewer 1 Report

Interesting paper. However, there are fundamental research design and methodological issues in this study.

Specific Points:

1. Please state the hypothesis.

2. Please include background information on the two magnetic therapies

3. Please explain why you consider testing the potential difference between the two magnetic applications essential.

4. Group 1 consisted of 6 men, while Group 2 consisted of 10 men. How did this happen? Please describe the group assignment process.

5. Nonetheless, this study should have been carried out as 3-arm RCT (with a no-intervention group). It is well-known that the bias associated with a non-random allocation could lead to consistent over- or underestimations of treatment effects.

6. “The study on group 1 was conducted in early spring and on group 2 in late autumn” Please state the exact dates. Also, have you considered the seasonal variations?

7. Y-axis should be standardized between Figures (1-4)

8. Pre-intervention circadian concentrations of serotonin appear distinctively different between Figures 1 and 2. Please discuss.

9. Figure 3 indicates a significant difference between the groups before applying magnetic fields. This is a serious problem.

10. Figure 4: I do not understand the point of comparing inadequately allocated two intervention groups.

11. Please add the Limitation statements in the Discussion section.

12. The data presented do not support the conclusion as stated.

Author Response

Dear Reviewer, 

Thank you very much for all your comments. They are very valuable. We made changes as suggested.

1.Please state the hypothesis. 

Hypothesis: Magnetotherapy and magnetic stimulation have an influence on serum serotonin concentrations in patients with low back pain.

2.Please include background information on the two magnetic therapies

We added it.

3.Please explain why you consider testing the potential difference between the two magnetic applications essential.

We added it.

4.Group 1 consisted of 6 men, while Group 2 consisted of 10 men. How did this happen? Please describe the group assignment process.

Patients could not be randomized due to the inclusion and exclusion conditions for the study. We managed to collect groups that were quantitatively sufficient for statistics. Patients were sent to rehabilitation with low back pain and if they were healthy and did not take medication, they could be included. There were really few such men. Most of the rehabilitation patients are people with more severe illnesses and disabilities.

5.Nonetheless, this study should have been carried out as 3-arm RCT (with a no-intervention group). It is well-known that the bias associated with a non-random allocation could lead to consistent over- or underestimations of treatment effects.

The absence of a control group was caused by a rather costly study program and a small number of men who could be enrolled in the study at a similar time (a significant equal length of day and night). It is planned to perform a study of a greater scope.

6.“The study on group 1 was conducted in early spring and on group 2 in late autumn” Please state the exact dates. Also, have you considered the seasonal variations?

The length of the day and night in this period of time was similar, the annual fluctuations in the secretion of such hormones as melatonin or serotonin are very high between winter and summer, which depend on the time of exposure and were taken into account. Blood samples were collected at night under red light, which did not disturb the secretion of hormones.

7.Y-axis should be standardized between Figures (1-4)

We tried to change that. The statistics program does not offer the possibility of chart standardization. 

8.Pre-intervention circadian concentrations of serotonin appear distinctively different between Figures 1 and 2. Please discuss.

Pre-intervention hormone levels could have varied, e.g. due to different pain levels in patients, being subject to individual biological cycles. Unfortunately, no pain charts were kept, as conclusions on this subject emerged after the results were obtained.

Good point. This requires further research.

9.Figure 3 indicates a significant difference between the groups before applying magnetic fields. This is a serious problem.

The explanation of this phenomenon is similar to that in the point above, all concentrations did not exceed physiological levels, presenting the natural differences that may occur in nature. In addition, the reaction after application is important, as it is regular and significant.

10.Figure 4: I do not understand the point of comparing inadequately allocated two intervention groups.

There is a clear tendency for a significant increase in serotonin concentration after magnetostimulation at time points at 8.00, 12.00, 20.00, 0.00, in which it was previously significantly lower. The increase in concentrations seems to be due to the effect of the field, which is also confirmed by other scientific studies quoted in the discussion. It can be seen from the concentration graph after magnetotherapy that it has not undergone any significant changes, i.e. the effect of this program was not marked.

11.Please add the Limitation statements in the Discussion section.

We added it.

12.The data presented do not support the conclusion as stated.

The final conclusions have been changed.

Reviewer 2 Report

The topic of this study is in the scope of the journal.

English language is appropriate throughout the manuscript, which is logical structured.

I am not sure, if the body height and body weight is important for the abstract, particular these are values from the population of this study. Please provide information on participants in abstract and method part too.

The introduction provides a good, generalized background of the topic that quickly gives the reader an appreciation of the research problem. It could be more fluent and consistent, with more “laser light” pointing main problem hypothesis.  

The objective is clearly defined in the last sentence introduction.

I think the discussion section of this study need to be made clearer. Please avoid bullets and numbering in discussion and conclusion section and numbers. I feel this is an important for this paper to make fluent and easy-reading discussion.  

No significant limitations are discussed.

Author Response

Dear Reviewer, 

Thank you very much for all your comments. They are very valuable. We made changes as suggested.

I am not sure, if the body height and body weight is important for the abstract, particular these are values from the population of this study. Please provide information on participants in abstract and method part too.

Patients admitted to rehabilitation at that time were examined in terms of weight and height and their BMI was within the normal range. They had to meet very strict conditions for inclusion, not taking any medications, and not having any endocrine or organic diseases. They differed from completely healthy men in the presence of low back pain.

The introduction provides a good, generalized background of the topic that quickly gives the reader an appreciation of the research problem. It could be more fluent and consistent, with more “laser light” pointing main problem hypothesis.  

We added it.

The objective is clearly defined in the last sentence introduction.

I think the discussion section of this study need to be made clearer. Please avoid bullets and numbering in discussion and conclusion section and numbers. I feel this is an important for this paper to make fluent and easy-reading discussion.  

We changed it as you suggested. 

No significant limitations are discussed.

We added it.

Round 2

Reviewer 1 Report

I appreciate the authors responding to my comments and making some improvements.

However, it became clear that this study was based on their clinical observation rather than comparative research.

This study's title should include "Clinical Observation Study." Please state clearly the type of research in the abstract and main text so that readers would not get confused thinking this was a Comparative study.

4.Group 1 consisted of 6 men, while Group 2 consisted of 10 men. How did this happen? Please describe the group assignment process.

>Patients could not be randomized due to the inclusion and exclusion conditions for the study. We managed to collect groups that were quantitatively sufficient for statistics. Patients were sent to rehabilitation with low back pain and if they were healthy and did not take medication, they could be included. There were really few such men. Most of the rehabilitation patients are people with more severe illnesses and disabilities.

I do not quite understand what the authors are stating. Nevertheless, this issue should be included in the Limitation section with clarity.

5.Nonetheless, this study should have been carried out as 3-arm RCT (with a no-intervention group). It is well-known that the bias associated with a non-random allocation could lead to consistent over- or underestimations of treatment effects.

>The absence of a control group was caused by a rather costly study program and a small number of men who could be enrolled in the study at a similar time (a significant equal length of day and night). It is planned to perform a study of a greater scope.

This issue should be included in the Limitation section with clarity.

7.Y-axis should be standardized between Figures (1-4)

>We tried to change that. The statistics program does not offer the possibility of chart standardization.

There are various methods available to create figures with the standardized y-axis.

8.Pre-intervention circadian concentrations of serotonin appear distinctively different between Figures 1 and 2. Please discuss.

>Pre-intervention hormone levels could have varied, e.g. due to different pain levels in patients, being subject to individual biological cycles. Unfortunately, no pain charts were kept, as conclusions on this subject emerged after the results were obtained. Good point. This requires further research.

This issue should be included in the Limitation section with clarity.

9.Figure 3 indicates a significant difference between the groups before applying magnetic fields. This is a serious problem.

>The explanation of this phenomenon is similar to that in the point above, all concentrations did not exceed physiological levels, presenting the natural differences that may occur in nature. In addition, the reaction after application is important, as it is regular and significant.

This issue should be included in the Limitation section with clarity.

10.Figure 4: I do not understand the point of comparing inadequately allocated two intervention groups.

>There is a clear tendency for a significant increase in serotonin concentration after magnetostimulation at time points at 8.00, 12.00, 20.00, 0.00, in which it was previously significantly lower. The increase in concentrations seems to be due to the effect of the field, which is also confirmed by other scientific studies quoted in the discussion. It can be seen from the concentration graph after magnetotherapy that it has not undergone any significant changes, i.e. the effect of this program was not marked.

"tendency" or "significant increase"? Which is it?

How do you know if the changes had anything to do with the interventions?

11.Please add the Limitation statements in the Discussion section.

>We added it.

This section should be expanded.

Author Response

Dear Reviewer,

thank you so much for your all comments. 

1. We add to the title - "Clinical Observation Study".

2. We added all of information in the Limitation section in Discussion. 

3. I use another program to create the figures, it's the same situation. 

Reviewer 2 Report

Dear Authors, 

Thank you very much for all your comments and i can see improvement in manuscript.

Best regards

Author Response

Dear Reviewer,

thank you so much.

Best regards,

This manuscript is a resubmission of an earlier submission. The following is a list of the peer review reports and author responses from that submission.